# Treatment with Manganese Porphyrin, MnTnBuOE-2-PyP^5+^, Suppressed the Activation of Macrophages in a Mouse Intracerebral Hemorrhage

**DOI:** 10.3390/ph18040547

**Published:** 2025-04-08

**Authors:** Shasha Zhang, Jie Cao, Ivan Spasojevic, Miriam Treggiari, Huaxin Sheng

**Affiliations:** 1Department of Anesthesiology, Duke University Medical Center, Durham, NC 27710, USA; shasha.zhang@duke.edu (S.Z.); miriam.treggiari@duke.edu (M.T.); 2Department of Anesthesiology, The 4th Hospital of Hebei Medical University, Shijiazhuang 050011, China; 3Pharmacokinetics and Pharmacodynamics Core, Duke Cancer Institute, Durham, NC 27710, USA; jie.cao@duke.edu (J.C.); ivan.spasojevic@duke.edu (I.S.); 4School of Forensic Medicine, Shanxi Medical University, Jinzhong 030600, China; 5Department of Medicine, Duke University Medical Center, Durham, NC 27710, USA

**Keywords:** manganese porphyrin, MnTnBuOE-2-PyP^5+^, ischemic stroke, hemorrhagic transformation, neurologic outcome mouse, macrophage, ferroptosis, BMX-001

## Abstract

**Background**: Manganese porphyrin, MnTnBuOE-2-PyP^5+^ (BMX-001), improves neurologic deficits in experimental ischemic stroke and has the potential to serve as an adjunct with thrombolysis or thrombectomy in stroke patients. In 10–30% of stroke patients following thrombolysis, the hemorrhagic transformation, associated with iron release, occurs. This study aimed to examine the neurologic outcome following the BMX-001 treatment in a mouse intracerebral hemorrhage (ICH) model with relevance to prospective ischemic stroke clinical trials. **Methods**: Twenty C57Bl6 mice were randomly assigned to groups after surgery and received vehicle or BMX-001 treatment immediately following stereotaxic left striatum collagenase injection. Post-ICH body weight, the Corner test, neurological deficit score, and Rotarod test were examined. Six sham surgery mice serve as a control group. At 72 h, the brain histological evaluation was performed, including hemorrhage size, Prussian blue staining, and the activation of macrophages. Data were collected by a researcher who was blind to groups. **Results**: No significant difference in body weight, neurological deficits, and hemorrhage size was found between groups. However, BMX-001 reduced the number of macrophages in the hemorrhagic area (48 ± 10 in vehicle, 33 ± 8 in BMX-001, *p* = 0.008) and the number of cells stained with Prussian blue—an indicator of iron released during hemorrhage (65 ± 22 in vehicle and 41 ± 15 in BMX-001, *p* = 0.027). **Conclusions**: The results support the safe use of BMX-001 in stroke patients in combination with thrombolysis or thrombectomy and, moreover, indicate the beneficial anti-inflammatory effect of BMX-001, alike to that previously reported in stroke studies of analogous, similarly redox-active, Mn porphyrins.

## 1. Introduction

Ischemic stroke is a common disease leading to high mortality and disability in the world. The incidence of ischemic stroke has seemingly increased due to the increased longevity [1,2,3,4]. Because of a loss of blood supply, the brain function in the respective area is instantaneously affected, leading to subsequent neurological and cognitive deficits [5,6,7]. In general, the primary treatment of the disease is to restore blood flow as soon as possible. The FDA approved the intravenous tissue plasminogen activator (tPA) in 1996, and alteplase is widely used in patients [8,9,10,11]. Tenecteplase, a longer half-life bioengineered variant of alteplase, has also been tested in clinical trials [12,13]. Thrombosis and thrombectomy have demonstrated efficacy in improving survival and reducing functional deficits [9,12,14,15]. However, delayed door-to-needle time and a complicated pathological mechanism led to 40–50% of stroke patients with post-treatment disability [16,17,18]. Therefore, it is necessary to develop a compound that would either preserve the brain tissue to allow for a longer time window until the blood flow gets restored or would rescue the brain tissue adjunct to thrombolysis or thrombectomy. Ischemic tissue becomes fragile, and hemorrhagic transformation frequently occurs, especially following thrombolysis [8,9,11]. Consequently, the candidate compound serving as an adjunct to two primary stroke treatments should ideally work under both ischemic and hemorrhagic conditions.

Cationic Mn (III) *N*-substituted pyridylporphyrins (Manganese porphyrins, MnPs) belong to a class of compounds initially developed as powerful superoxide dismutase (SOD) mimics. Mn (III) *meso*-tetrakis (*N*-ethylpyridinium-2-yl)porphyrin (MnTE-2-PyP^5+^) was the first drug to exhibit the neuroprotective effect with a delayed treatment in an animal model of ischemic stroke [19]. Subsequent studies showed the redox ability of analogous Mn porphyrins to modify the activity of transcription factors, such as nuclear factor κB (NF-κB) with subsequent reduction in levels of proinflammatory cytokines in ischemic brain tissues [20,21]. The lead compound, Mn(III) *meso*-tetrakis(*N*-(2’-n-butoxyethyl)pyridinium-2-yl)porphyrin (MnBuOE-2-PyP^5+^, BMX-001), is currently tested in multiple clinical trials on high-grade glioma, multiple brain metastases, head and neck cancer, anal cancer, and rectal cancer [22]. Our preclinical studies have also demonstrated its protection against ischemic stroke [23], which motivated us to develop it for clinical trials in stroke patients.

Intracerebral hemorrhage could cause the accumulation of iron in the brain tissue [24,25,26], which would contribute to ferroptosis, a type of cell death that is distinct from the apoptosis [27,28,29]. A ferroptosis inhibitor has been used to treat t-PA-induced hemorrhagic transformation in ischemic stroke [30]. The activation of the Nrf2 pathway may play a role in suppressing ferroptosis, as it leads to an increase in the levels of endogenous enzymes that remove H_2_O_2_, such as catalase, glutathione peroxidase, and peroxiredoxins. This, in turn, could avoid the Fenton reaction between H_2_O_2_ and iron [31,32]. The safety of manganese porphyrin with relevance to prospective clinical stroke studies on patients that would receive BMX-001 treatment when hemorrhagic transformation occurs has not yet been addressed. Our work aims to study the intracerebral hemorrhage (ICH) in a mouse model, ensuring both the safe and efficacious use of BMX-001 in clinical trials of ischemic stroke.

## 2. Results

### 2.1. BMX-001 Does Not Worsen the Clinical Outcome in a Mouse Intracerebral Hemorrhage

Two mice were excluded from this study due to surgery-related death. Three mice, two vehicle mice, and one BMX-001-treated mouse, died after 24 h due to severe ICH-induced brain damage. The survival rates were 80% and 87.5% in the vehicle and BMX-001 treatment groups, respectively. All sham mice that underwent surgery survived.

After surgery, the mice had a body weight loss of around 2 g (25.0 ± 2.0 g at post-ICH 24 h vs. 26.9 ± 1.9 g pre-surgery, *p* = 0.017). The collagenase induced hemorrhage in the left striatum area, and the subsequently induced local injury did not cause further body weight loss at 48 and 72 h post-ICH (Figure 1A). However, these mice did have significant functional deficits. They turned to the side that is ipsilateral to the hemorrhage hemisphere. The Corner test was used to measure the turning behavior. Left striatum ICH increased the turns from 47.5 ± 28.1% to 81.2 ± 37.2% at 24 h post-ICH, *p* = 0.03 (Figure 1B). Rotarod was used for testing motor function. Hemorrhage reduced the Rotarod performance from 300 ± 0 s to 180 ± 119 s at 24 h post-ICH, *p* = 0.03 (Figure 1C). Both the Corner test and Rotarod performance were improved at 72 h post-ICH. No difference between vehicle and BMX-001 groups was found (*p* > 0.05). The mice also had neurological deficit score at 24 h post-ICH (Figure 1D, Vehicle 4.8 ± 2.9, BMX-001 5.1 ± 2.5, *p* = 0.9), and 72 h (Vehicle 2.8 ± 1.6, BMX-001 ± 3.0 ± 1.5, *p* = 0.8). The neurological deficits improved from 24 h to 72 h post-ICH (*p* = 0.019), and there was no difference between the vehicle and BMX-001 groups (*p* > 0.05).

The intracerebral hemorrhage was measured at 72 h post-ICH. The vehicle mice had a hemorrhage area of 11.83 ± 3.96 mm^2^, and the BMX-001-treated mice had a hemorrhage area of 10.87 ± 4.18 mm^2^ (Figure 2, *p* = 0.65). BMX-001 treatment did not enlarge the hemorrhage, although it could dilate the vessels.

### 2.2. BMX-001 Treatment Reduces Level of Macrophages in Mouse Intracerebral Hemorrhage

Many multinucleated giant cells were found in the area surrounding the hemorrhage (Figure 3). These large phagocytic cells are involved in cleaning debris and damaged cells from the hemorrhagic area. In BMX-001-treated brains, the number of these cells was reduced (33 ± 8 in the BMX-001 group and 48 ± 10 in the Vehicle group, *p* = 0.0079).

Prussian blue staining is a histology technique used for detecting iron in brain tissue following hemorrhage. Hemosiderin, an iron storage complex, can be found in macrophages. Prussian blue-stained cells were counted in the brain slides of surviving mice (Figure 4). BMX-001 treatment reduced the number of cells bearing iron at 72 h post-ICH as well (40 ± 14 in the BMX-001 treatment group, 65 ± 22 in the Vehicle group, *p* = 0.027).

## 3. Discussion

Thrombolysis and/or thrombectomy have been the standard of care in ischemic stroke patients. These treatments restore blood flow in the ischemic area, which is critical for stroke patients to achieve better outcomes. Based on the clinical report, complete recanalization was achieved only in 56.5% of thrombolysis patients [33], and the outcomes are tightly associated with the door-to-needle time for patients to receive the drug [34]. In the SKIP randomized clinical trial, Suzuki et al. reported that a favorable outcome occurred in 59.4% of stroke patients with mechanical thrombectomy and in 57.3% of stroke patients with intravenous thrombolysis plus mechanical thrombectomy [35]. As post-stroke functional deficits still remain a major concern, the compound that could be safely delivered adjunct to thrombolysis or thrombectomy would largely improve the stroke outcome.

Hemorrhagic transformation is a serious complication following thrombolysis in ischemic stroke patients. Clark et al. reported that t-PA treatment significantly increased the asymptomatic and symptomatic ICH rate up to 11.4% (*p* = 0.004 vs. placebo at 4.7%) [8]. Hacke et al. reported that thrombolysis, with alteplase given 3 to 4.5 h after acute ischemic stroke, had the incidence of ICH in 27% of patients (*p* = 0.001 vs. placebo at 17.6%). Hemorrhage also occurred in stroke patients following thrombectomy [36]. Thus, it is imperative that the compound, when delivered adjunct to thrombolysis or thrombectomy, is not harmful but efficacious to the brain when hemorrhagic transformation occurs.

BMX-001 is the lead manganese porphyrin presently in multiple clinical trials where its ability to protect normal tissue, while suppressing cancer growth, has been evaluated. Importantly, its low toxicity and high lipophilicity allows it to cross the blood–brain barrier [22] and makes it a preferred compound for treating neurological disorders. Our preclinical stroke experiments demonstrated its neuroprotective effect in the male and female, young and aged, as well as in rats with spontaneous hypertension [23]. BMX-001 was given through the carotid artery after the filament removal, which mimics the adjunct treatment with thrombectomy. Neurological function was consistently improved at 28 days post-stroke. The current study was intended to assess whether the BMX-001 is safe when hemorrhagic transformation occurs in ischemic stroke patients. Superoxide is formed mainly in ischemic brain [37]. Treatment with three analogous Mn (III) *N*-substituted pyridyl porphyrin-based SOD mimics of similar redox abilities, MnTE-2-PyP^5+^ (BMX-010, AEOL10113), MnTDE-2-ImP^5+^ (AEOL10150), and MnTnBuOE-2-PyP^5+^ (BMX-001), improved the neurological outcome in part through their ability to dismute the superoxide [19,20,21,23,38]. During superoxide dismutation, hydrogen peroxide (H_2_O_2_) is produced. If insufficiently taken care of, H_2_O_2_ could possibly react with iron via Fenton chemistry and in turn participate in the pathological process of ferroptosis [39]. Ferroptosis plays an important role in ICH and has been considered a new therapeutic target [27].

The current study has demonstrated that BMX-001 treatment did not worsen the neurological outcome and hemorrhagic size in mouse ICH. BMX-001-treated mice had a better survival rate compared to the vehicle. More importantly, BMX-001 treatment reduced the number of macrophages including iron-bearing cells during mouse ICH, which was supported by the cell counting/staining of multinucleated giant cells including Prussian blue positively stained cells. Multinucleated giant cells were found in the peri-hemorrhage area where they are involved in iron clearance; similar findings have also been reported by another group [40,41]. Therefore, the superoxide dismutation by BMX-001 and subsequent H_2_O_2_ production is not likely to be involved in ferroptosis. Our previous studies on ischemic stroke provided ample evidence that three similarly redox-active Mn porphyrins (listed above) reduced NF-κB activation with a subsequent reduction in levels of TNF-α and IL-6, thereby suppressing inflammatory processes [20,21,23]. The BMX-001-driven reduction in the number of macrophages and iron-bearing cells suggests the suppression of inflammation. The anti-inflammatory processes, including the inhibition of NF-κB activation and the neuroprotective role of Nrf2 signaling, will be further investigated. The limitation of this study is that it is an acute short-term experiment with a single-dose treatment. Future studies will involve long-term outcomes with multiple treatments to address these limitations and improve our understanding of the underlying mechanism.

## 4. Materials and Methods

### 4.1. Animal Model and Housing

The following experiments were approved by the Duke University Institute Committee of Animal Use and Care. All animal handling and care were in accordance with the National Institutes of Health guidelines. Twenty male C57BL/6 mice (8–12 weeks old, weighing 24–32 g) were purchased from Jackson Laboratory (strain # 000664, Bar Harbor, ME, USA). Mice were housed at Duke vivarium under standard conditions, with free access to food and water and a 12 h light/12 h dark cycle. Room temperature and humidity were well controlled.

### 4.2. Intracerebral Hemorrhage (ICH) Model and Drug Treatment

The ICH model was induced under 3% isoflurane inhalation for anesthesia induction and 1.5% for maintenance in 40% oxygen balanced with nitrogen. Body temperature was maintained through the water-circulated plate with the temperature of 38 °C. The mice were kept in the prone position, and lubricant eye ointment was applied to both eyes. The surgical area was shaved and cleaned with iodine and alcohol. A small skin incision was cut, and a burr hole was made in the left skull at 0.5 mm anterior, 2.4 mm lateral to bregma. A stereotaxic injection of 0.3 μL saline containing 0.0225 U of clostridial collagenase (Type IV-S, Sigma, St. Louis, MO, USA) was delivered into the left striatum (2.8 mm below the cortical surface) using a pulled glass needle at a rate of 80 nL/min (Syringe pump, WPI SMARTouch^TM^, Sarasota, FL, USA). The needle remained at the site for 10 min and then was removed. The burr hole was sealed with bone wax (CPB31A, CP Medical, Norcross, GA, USA), and the skin incision was closed, and an antibiotic ointment was applied to the skin surface. The isoflurane was turned off to allow mice to recover at room temperature. The mice were randomly assigned to the groups (vehicle *n* = 11 and BMX001 *n* = 9). They received a single intraperitoneal injection of BMX-001 (450 μg/kg) or vehicle (same amount of 0.9% Sodium chloride, Hospira, Lake Forest, IL, USA) at one hour post-surgery. The sham-operated mice (*n* = 6) received the surgical procedure and had no collagenase injection.

### 4.3. Neurological Functional Assessment

Corner Test: The apparatus consisted of two opaque plexiglass boards (30 × 20 cm, 5 mm thick) fixed at a 30° angle, with small openings (~0.9 cm at the top and ~0.4 cm at the bottom) to guide mice into the corner [42]. Each mouse was placed individually between the boards, facing the corner, with its back to the handler. Care was taken to avoid unilateral whisker contact or excessive handling, which could influence behavior. A turn was recorded as complete if the mouse touched both boards with its vibrissae, turned 180°, or rotated its head at an angle exceeding 90°. Each mouse underwent 10 trials, and the percentage of left turns was calculated as the number of left turns divided by 10 × 100%.

Rotarod Test: Mice were habituated to the rotarod apparatus (Ugo Basile, Gemonio, Italy) for three days prior to testing. During the test, the rotation speed was gradually increased from 4 RPM to 40 RPM over 300 s [43]. Each mouse underwent three trials at 15 min intervals, and the latency to fall was recorded. The best performance of three trials was used for analysis.

Neurological deficits were evaluated on days 1 and 3 post-ICH. Assessments included spontaneous activity, circling behavior, limb movement symmetry, forepaw extension, body sensation, whisker sensation, facial sensation, beam walking, and grid climbing. A score of 0 to 4 was given in each category, and 0 represents normal behavior [44]. The sum of the scores served as the neurologic deficit score for each animal.

### 4.4. Histological Analysis

Mice were euthanized under deep isoflurane anesthesia, followed by cardiac perfusion with 0.9% normal saline and 4% paraformaldehyde (PFA). Brains were post-fixed in 4% PFA overnight, paraffin-embedded, and sectioned coronally through the entire lesion area at a thickness of 5 μm with an interval of 500 μm between sections. Several sets of sections were collected according to the experimental design.

Hematoxylin and Eosin (H&E) Staining: Paraffin-embedded brain sections were deparaffinized in xylene (2 min, 2 changes), rehydrated through graded ethanol (100%, 95%; 2 min each, 2 changes per concentration), and rinsed in distilled water (2 min per change). Sections were stained with Harris Hematoxylin (4 min), rinsed in distilled water (3 changes, 10 dips per change), differentiated in 10% acid alcohol (2 min), and rinsed again in distilled water (3 changes, 10 dips per change). Sections were blued using 3% ammonia water (15 s), followed by rinsing in distilled water (3 changes, 10 dips per change). Counterstaining was performed with eosin (2 min), followed by dehydration in graded ethanol (95%, 3 min, 3 changes; 100%, 5 min, 3 changes), clearing in xylene (5 min, 3 changes), and mounting with Cytoseal XYL (Thermo Scientific, REF: 8312-4, Waltham, MA, USA).

Prussian Blue Iron Staining: To detect iron deposits in brain tissue, Prussian Blue staining was performed using a commercial kit (ab150674, Abcam, Cambridge, UK). Paraffin sections were deparaffinized in xylene, rehydrated through graded ethanol (100%, 95%, 70%), and rinsed in distilled water. A working solution, prepared by mixing equal volumes of potassium ferrocyanide and hydrochloric acid, was applied to the sections for 3 min. Following rinsing in distilled water, sections were counterstained with Nuclear Fast Red (5 min), rinsed in distilled water (4 changes), dehydrated in 95% ethanol and absolute ethanol, cleared in xylene, and mounted with synthetic resin. Images of the entire brain were captured at 20 × magnification to visualize iron deposits.

Hemorrhage size measurement and cell counting: Sections with the largest hemorrhagic area were captured and analyzed using ImageJ software (Prism 10, the 10th version, NIH, Bethesda, MD, USA) to quantify the hemorrhagic region. The numbers of multinucleated giant cells and the cells with Prussian blue were counted from the entire area of the lesion in the H&E or Prussian blue-stained brain sections, respectively.

All behavioral tests and histological data collection were performed by an investigator blind to the treatment groups.

### 4.5. Statistical Analysis

Data are presented as mean ± standard deviation (SD), unless otherwise noted. We utilized two-way analysis of variance (ANOVA) for assessing body weight, Corner test performance, and Rotarod performance. For histological analysis, we employed an unpaired Student’s *t*-test. Neurological deficit scores, as nonparametric data, are summarized as median ± interquartile range (IQR) and analyzed using the Mann–Whitney test. All statistical analyses were conducted with Prism 10 software (GraphPad Software Inc., San Diego, CA, USA), and a *p*-value of less than 0.05 was considered statistically significant.

## 5. Conclusions

The current study has demonstrated that BMX-001 treatment in ischemic stroke is safe when delivered adjunct to thrombolysis or thrombectomy. Further, in the event of hemorrhagic transformation, BMX-001 did not worsen the neurological outcome in a mouse intracerebral hemorrhage. Moreover, the reduction in the number of macrophages and iron-bearing cells by BMX-001 in the hemorrhagic area suggests its anti-inflammatory mode of action.

## 6. Patents

No patent has resulted from the work reported in this manuscript.

## Figures and Tables

**Figure 1 pharmaceuticals-18-00547-f001:**
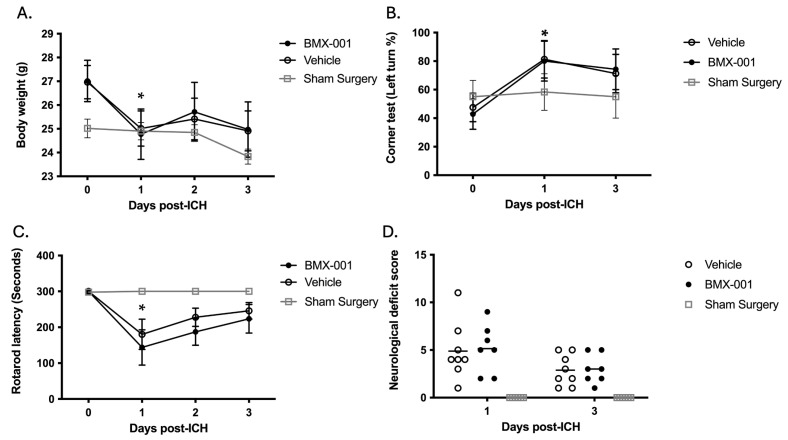
Body weight and neurological functional changes in mouse intracerebral hemorrhage: (**A**) Body weight. (**B**) Corner test. (**C**) Rotarod latency. (**D**) Neurological deficit score. *n* = 8 in the Vehicle group, 7 in the BMX-001 group, and 6 in the sham surgery group. Sham surgery mice serve as a reference group and are not intended for statistical analysis. Mean ± SD, * *p* < 0.05 vs. day 0.

**Figure 2 pharmaceuticals-18-00547-f002:**
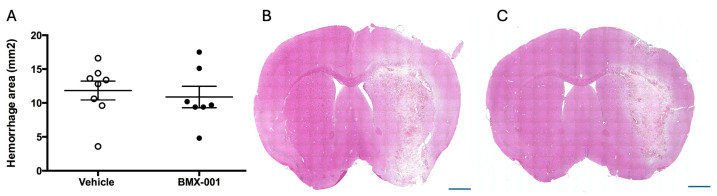
Intracerebral hemorrhage size measured at 72 h post-surgery: (**A**) Hemorrhage area. (**B**) H&E-stained vehicle brain. (**C**) H&E-stained BMX-001-treated brain. *n* = 8 in the Vehicle group, and 7 in the BMX-001 treatment group. Mean ± SD, Blue Bar = 1 mm.

**Figure 3 pharmaceuticals-18-00547-f003:**
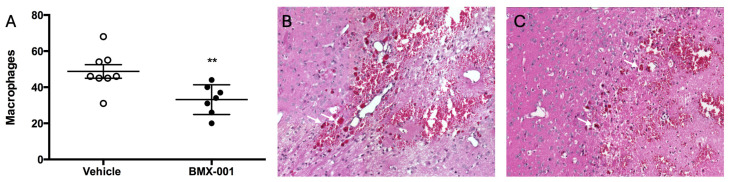
Multinucleated giant macrophage cells in brain at 72 h post-ICH: (**A**) The number of this type of macrophages in both vehicle and BMX-001 treatment groups. (**B**) H&E-stained vehicle brain. (**C**) H&E-stained BMX-001 brain. *n* = 8 in the Vehicle group and 7 in the BMX-001 group. Mean ± SD, ** *p* < 0.01, the white arrow points to the multinucleated giant cell.

**Figure 4 pharmaceuticals-18-00547-f004:**
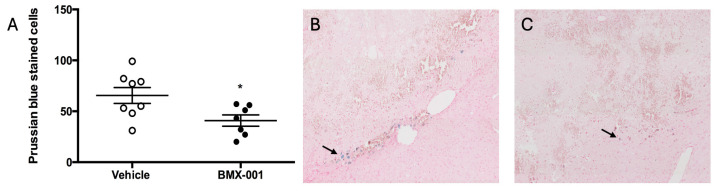
Prussian blue-stained cells at 72 h in post-ICH brains: (**A**) The number of these cells in both vehicle and BMX-001 treatment groups. (**B**) Vehicle brain. (**C**) BMX-001-treated brain. *n* = 8 in the vehicle and 7 in the BMX-001 treatment, Mean ± SD, * *p* < 0.05, the black arrow points to the Prussian blue positive cells.

## Data Availability

All data were stored in the University LabArchives eResearch notebook and will be available upon request.

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
