# Peer review of "Treatment with Manganese Porphyrin, MnTnBuOE-2-PyP5+, Suppressed the Activation of Macrophages in a Mouse Intracerebral Hemorrhage"

_pharmaceuticals, 2025, doi:10.3390/ph18040547_

Round 1

Reviewer 1 Report

Comments and Suggestions for Authors

excellent work,

minor commmenst related tominor Englsih editing

Author Response

Thank you for your review. We have checked the English.

Reviewer 2 Report

Comments and Suggestions for Authors

In the present manuscript, the authors described the results of a series of experiment performed in an animal model of ICH. They demonstrated that the treatment of mice with manganese porphyrin, MnTnBuOE-2-PyP5+ (BMX-001), did not affect neurological deficit and the lesion size, but decreased the number of multinuclear macrophages and iron-containing cells in the damaged area. This study is poorly designed and conducted. 

1. The authors did not include any control group in the study. The studies of this type should include at least one sham-operated group. In the present protocol, it is not possible to understand what is normal level of any indices studied. 

2. There is not any information how the histological data were obtained and treated. The authors should present how many sections were prepared and stained from each animal, how it was related to the size of lesion, what area was chosen for analyses, how the cells were counted, how the lesion area was measured. 

3. Complete time-table (scale) of all behavioral tests should be presented. Were the animals handled before surgery?

4. There is not any description of statistical analysis. It is not possible to understand how these "significant" differences were calculated.

5. The authors used only one dose of the drug. How the dose was chosen?  

6. The final conclusion: "The results support the safe use of BMX-001 in stroke patients in combination with thrombosis or thrombectomy and, moreover, indicate the beneficial anti-inflammatory effect of BMX-001, alike that previously reported in stroke studies of analogous, similarly redox-active, Mn porphyrins." seems to be senseless because there were not any experiments directed to study inflammatory indices in the paper. 

Author Response

Thank you for your insightful comments. Please find our attached responses for your review.

Reviewer 3 Report

Comments and Suggestions for Authors

The article titled "Treatment with Manganese Porphyrin, MnTnBuOE-2-PyP5+, Suppressed the Activation of Macrophages in a Mouse Intracerebral Hemorrhage" by Zhang et al. presents an insightful evaluation of the compound BMX-001 as a potential adjunct treatment for stroke patients undergoing thrombolysis or thrombectomy. The preliminary results suggest that the administration of this agent is safe; however, it does not appear to provide additional benefits in terms of neurological outcomes in the examined mouse model.

Specific Comments:

Line 5: The full name of the first author is missing.

Line 31: The term "thrombosis" is used; however, "thrombolysis" may be the intended term. Please clarify as needed.

Lines 67-73: The content in this section could be presented more coherently to improve flow and readability.

Line 73: The term "Fenton chemistry" should be replaced with "Fenton reaction" for accuracy. Any other instances of this terminology should also be corrected accordingly.

Results Section:

The results section is somewhat disorganized, making comprehension challenging. Certain methodological details appear in this section when they are more fitting in the methods section.

Discussion Section:

The discussion could be expanded and refined to provide a more comprehensive analysis. Specifically:

1) Among the various BMX analogs, why was BMX-001 selected? What advantages does it offer over other analogs?

2) Could additional tests or stains have been conducted to better assess ferroptosis in the samples?

3) Are there alternative assays that could have concurrently evaluated NF-κB activation and levels of TNF-α, IL-6, and other relevant inflammatory markers in the samples?

4) Given the observed significant changes in macrophage numbers and iron release in the treatment group, how can the lack of differences in neurological deficits between groups be explained?

5)The findings suggest that while BMX-001 appears to be safe, it does not confer any substantial clinical benefits. A discussion of potential reasons for this observation would be valuable.

6) What are the key strengths of this study, and what limitations should be acknowledged?

Author Response

Thank you so much for your insightful comments. Please find our attached responses for your review.

Round 2

Reviewer 2 Report

Comments and Suggestions for Authors

The authors improved their manuscript. However, some aspects remained unclear.

  1. In fig.1A, body weight of the sham-operated group was significantly lower than of the experimental groups. This shows inappropriate experimental design.
  2. The authors should present any information about the volume, not area of lesion, because this is the index, which determines the other measured parameters. Brain is not a 2D plate, this is a 3D organ.
  3. I did not understand the description of cell count presented by the authors. It is quite inappropriate. What does it mean: "The numbers of multinucleated giant cells and the cells 268 with Prussian blue were manually counted from the entirety of the lesion, rather than 269
    from a single field of view, in the respective H&E or Prussian blue stained brain sections"? How it was performed? In one section next to the H&E-stained section? What does it mean "entire" in this case? I suggest the authors to present these data at least as cell density with clear identification that this was performed in one section with a maximum size of lesion. The area of counting should be clearly shown in photos. 
Comments on the Quality of English Language

Quality of English Language seems to be suitable to understand main data but may be improved.

Author Response

Comment #1. In fig.1A, body weight of the sham-operated group was significantly lower than of the experimental groups. This shows inappropriate experimental design.

Response #1. We had to meet the resubmission deadline, so we did not wait for the body weight to match that of the injured group. The purpose of adding this sham-operated group is to serve as a reference for changes in functional deficits after surgery.   

Comment #2. The authors should present any information about the volume, not area of lesion, because this is the index, which determines the other measured parameters. Brain is not a 2D plate, this is a 3D organ.

Response #2. In the literature, selecting the level of the largest injury area for data analysis is common. We employed the same method in this experiment.  

Comment #3. I did not understand the description of cell count presented by the authors. It is quite inappropriate. What does it mean: "The numbers of multinucleated giant cells and the cells with Prussian blue were manually counted from the entirety of the lesion, rather than from a single field of view, in the respective H&E or Prussian blue stained brain sections"? How it was performed? In one section next to the H&E-stained section? What does it mean "entire" in this case? I suggest the authors to present these data at least as cell density with clear identification that this was performed in one section with a maximum size of lesion. The area of counting should be clearly shown in photos. 

Response #3. The lesion is large and cannot be observed in a single field of view. We need to move from one field to another to count the number of multinucleated giant cells on the H&E-stained slides, as well as the number of cells on the Prussian blue-stained slides, throughout the entire area of the lesion. We have changed the sentences in the manuscript.

Reviewer 3 Report

Comments and Suggestions for Authors

The authors have sufficiently addressed the previous comments. I have no further remarks.

Author Response

Thank you for your support

Round 3

Reviewer 2 Report

Comments and Suggestions for Authors

Unfortunately, the authors corrected their manuscript only insignificantly. They did not follow any advices made in the previous comment. 

Sham-operated animals of the other age and body weight were included that is not acceptable. 

Cell count in the present form is completely incorrect.